fluid mechanics/microsystems

capillary bridge, capillary force, evaporation

**Author for correspondence:**
Loïc Tadrist
e-mail: loic.tadrist@uliege.be

# Characterization of interface properties of fluids by evaporation of a capillary bridge

Loïc Tadrist[1], L. Motte[2], O. Rahli[2] and Lourès Tadrist[2]

[1]Microfluidics Lab, Department of Mechanical and Aerospace Engineering, University of Liege, Allée de la découverte 9, Liège 4000 Belgium
[2]CNRS, Institut Universitaire des Systèmes Thermiques et Industriels UMR 7343, Aix-Marseille Université, Marseille 13453 France

LT, 0000-0001-6200-9842

The surface properties between two non-miscible fluids are key elements to understand mass transfer, chemistry and biochemistry at interfaces. In this paper, surface properties are investigated in evaporating and non-evaporating conditions. A capillary bridge between two large plates (similar to a Hele-Shaw cell) is considered. The temporal evolution of surface forces and mass transfers due to evaporation of the liquid are measured. The force depends on surface properties of the substrate. It is adhesive in the wetting case and repulsive in the non-wetting case. The force is also shown to depend linearly on the volume of the capillary bridge $F \propto V_0$ and inversely to the height of the bridge. Modelling is performed to characterize both surface force and evaporation properties of the capillary bridge. The evaporation is shown to be diffusion driven and is decoupled from the bridge mechanics.

## 1. Introduction

Interface phenomena have been extensively studied since the seminal works of Plateau [1] and Rayleigh [2]. The shape of a liquid interface is easily described by the Young–Laplace equation. However, this equation is not easy to solve even in the case of an axi-symmetric liquid bridge between two parallel plates, which is one of the simplest cases to study interfaces. This classroom case has given birth to an abundant bibliography mixing theoretical works as well as experimental ones [3].

Those studies were carried out to understand various phenomena; from the most theoretical ones with the onduloid shapes with null pressure [4], or the break-up of a liquid bridge [5]; to the most applied ones such as the building of a sand castle [6]. From a biological point of view, capillary bridges are

**Figure 1.** Schematics of the experimental bench. The profile of the drying liquid bridge is recorded by a camera. The force of the liquid bridge is measured by a precision scale. A pole gives vertical control on the upper plate and allows us to fix the height $z_1$ of the bridge.

also present in the adhesion of insects [7] for instance. The complex cases of the evaporation of a capillary bridge between one sphere and a surface [8] or between two spheres have been studied considering both evaporation kinetics [9] and bridge break-up [10]. Recently, the evaporation of a capillary bridge between two plates has been extensively explored in well-controlled conditions by Portuguez *et al.* [11,12] for different wetting angles and air humidities. However, capillary forces of evaporating liquid bridges were not directly measured with their set-up.

The capillary bridge is one of the most standard configurations in which interface effects may be studied and characterized dynamically for liquids. It differs from the evaporation of a sessile drop in three main aspects. First, the control of the height of the bridge $z_1$ allows us to start the drying with receding contact angles directly, whereas in the case of a sessile drop the contact angle may vary from advancing to receding at the beginning of the drying. Second, the control of the height of the bridge also allows us to control finely the evaporation kinetics of the liquid. Finally, the capillary force might be measured which gives a direct insight on the evolution of the surface properties. This differs from the case of a sessile drop for which this quantity is not easily accessible. Those three aspects make the capillary bridge technique more robust than the sessile drop technique to characterize the surface properties of fluids.

Bacteria and other microbes thrive at the liquid–gas interface, mainly thanks to the easy access to oxygen from the gas and nutrients from the liquid. Some bacteria secrete biological molecules (enzymes, proteins, etc.) that may change the bulk and interface properties of the liquid. Recent studies have shown that both the evaporation rate [13] and the surface tension [14] of specific bacteria solutions can be changed by one order of magnitude. Those bio-surfactants or bio-films may be industrially used not only to produce pharmaceutical products [15] but also to process food (dairy transformations [16], specific wine process [17], etc.). In this context several attempts have been made to characterize the effect of bio-secretions on interface properties, such as the drop-collapse test [18].

We focus here on the dynamics of an evaporating capillary bridge between two large plates to finely control the boundary conditions for evaporation. We aim at characterizing the evaporation rate and the surface forces generated by the capillary bridge. We demonstrate that our experiment is efficient at measuring the surface parameters (evaporation rate and surface forces) of any volatile liquid.

The paper is organized as follows. Section 2 is devoted to the description of the experimental set-up, measurement techniques and experimental procedure. We then present the experimental results obtained with our set-up in §3. Finally, those results are compared to analytical models presented in §4. Those models are based on (i) the Young–Laplace equation for the force of the capillary bridge and (ii) mass transfer equation for evaporation rate.

## 2. Material and methods

### 2.1. Experimental set-up

The experiment is depicted in figure 1. It is constituted of two horizontal flat plates in the centre of which a liquid drop of a desired initial volume $V_0$ is deposited. The two plates are flat cylinders of radius $L = 2$

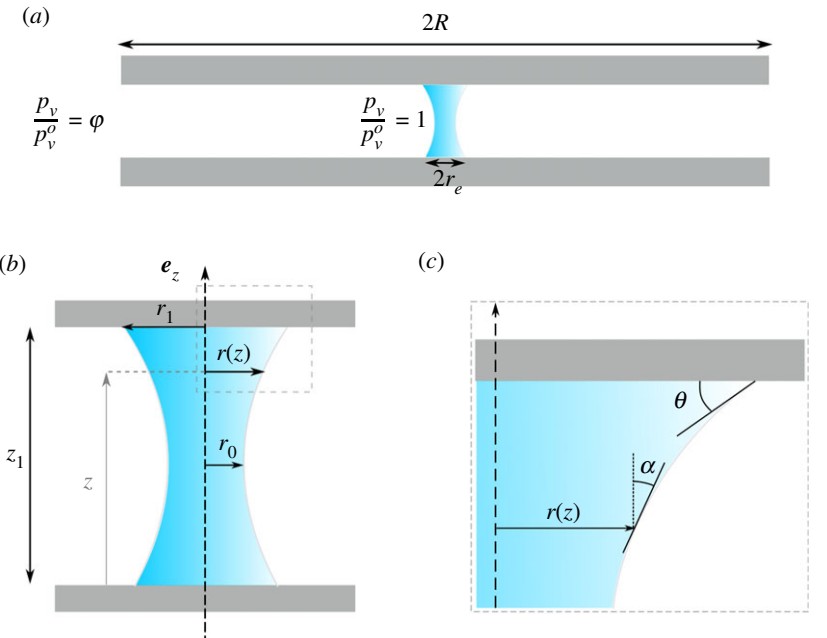

**Figure 2.** Schematics of the liquid bridge and definition of parameters used throughout the paper. (*a*) Schematics of the drying of a liquid bridge: close to the liquid–air interface, air is saturated by water vapour, $p_v/p_v^o = 1$. Away from the two substrate plates, air has a constant water vapour saturation $p_v/p_v^o = \varphi$. (*b*) Schematics of a liquid bridge on a hydrophilic substrate having a critical point in $r_0$. The liquid bridge is described by the radius $r(z)$ and by its contact angle $\theta$. (*c*) Close-up on the liquid–gas and the definition of angle $\alpha$ with $\cos \alpha = 1/\sqrt{1 + \dot{r}^2}$.

and 0.4 cm thick made of PVC (mass 7g). The upper plate is lowered, thanks to a vertical elevator, until the drop bridges the two plates. By back-moving the upper plate, we simply fix the height $z_1$ of the capillary bridge.

To avoid any contamination of the surfaces, the PVC plates are cleaned with ethanol and dried before each measurement. The lower plate is placed onto the precision scale and a distilled water drop of volume $V_0$ is deposited on its centre. The precision scale (Mettle Toledo xs205 with a precision of ±1 μN) is balanced and set to 0. The upper plate is then approached to create the liquid bridge. With such a procedure, the precision scale only weighs the effect of the capillary bridge. Indeed, as soon as the liquid bridges the two plates, the scale shows a negative weight due to the capillary adhesion for a wetting fluid. Conversely, the scale shows a positive weight corresponding to a repulsion force for a non-wetting liquid. In order to investigate the effect of the surface properties of the substrate, a thin layer of the considered material (glass, Teflon or aluminium) is stuck to the PVC plate and the same procedure is applied. The experiments were carried out at an ambient temperature ($T = 27°C$) and pressure ($P = 1015$ hPa). The room humidity ($\varphi \sim 0.65 \pm 0.05$) was measured for each evaporation test.

Along with the force measurement, we record the shape of the liquid bridge using a camera placed on the side. We check the image if the two plates are horizontal by measuring the distance between the two plates edges. The images are automatically processed using image processing tools from Matlab. The shape descriptors of the liquid bridge ($r(z)$ and $\dot{r}(z) = \partial r/\partial z$) are computed on each image. From those measurements, we deduce the surface $S$ and volume $V$ by

$$S = \int_0^{z_1} 2\pi r \sqrt{1 + \dot{r}^2}\, dz \quad \text{and} \quad V = \int_0^{z_1} \pi r^2\, dz. \tag{2.1}$$

Finally, we obtain the simultaneous time evolution of the capillary force $F(t)$, volume $V(t)$ and surface $S(t)$ of the capillary bridge during the whole drying process until break-up.

## 2.2. Observables and notations

The parameters used in the paper are defined in figure 2. The profile of the liquid bridge $r(z)$ ends with a contact angle $\theta$ on the plate. For simplicity, we denote $r_0$ as the radius of the liquid bridge at the neck of

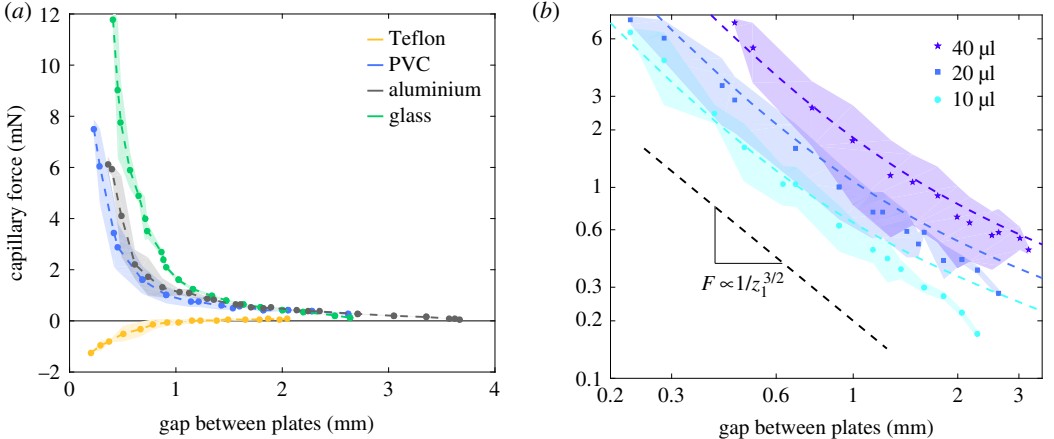

**Figure 3.** Variation of the force $F$ of a liquid bridge of fixed volume as a function of bridge height $z_1$. (a) For wetting surfaces glass (green), PVC (blue) and aluminium (grey), the force is purely attractive and decreases with $z_1$ (drop volume $V_0 = 20$ µl). However, in the case of Teflon (yellow), a non-wetting surface, the force is initially repulsive before getting almost null. (b) The log–log plot of the variation of the force $F$ of a water–PVC liquid bridge of different volume (filled circle) $V_0 = 10$ µl, (filled square) $V_0 = 20$ µl and (star) $V_0 = 40$ µl. The force decreases following a power law $F \propto z_1^{-3/2}$. The coloured dashed lines correspond to fits using equation (4.8) and $\theta = 80°$.

the bridge (or equivalently at the point of the largest radius of the bridge if the curvature along $z$ is negative) and $r_1$ the radius of the liquid bridge in contact with the plate.

# 3. Experimental results

We start by performing tests on capillary bridges to study the effect of drying on the adhesion force. The first step is to measure the effect of geometry and substrate properties on the force generated by a capillary bridge in negligible evaporation conditions. The second step is to measure the force of a capillary bridge when evaporation occurs. Finally, we characterize the evaporation regime.

## 3.1. Force of a capillary bridge

We first varied the height $z_1$ of the liquid bridge and recorded the force $F$ with a fixed volume $V_0$ (figure 3). The time scale of this experiment is less than 200 s. As shown in figure 4a, the loss of liquid due to evaporation within this time scale is negligible. For each test the experiment has been repeated three times and the experimental points of figure 3a,b correspond to the average of the three experiments. The coloured shadow corresponds to the envelope of the uncertainties.

The capillary force is attractive in the case of wetting substrates (glass, aluminium and PVC) and repulsive in the case of a non-wetting substrate (Teflon) (figure 3a). We observe that the force is maximal in magnitude when the two plates are the closest. The force generated by the capillary bridge of a fixed volume decreases in magnitude with the height of the capillary bridge. For the wetting substrates, the capillary force decreases smoothly until the bridge breaks up. In the case of the non-wetting substrate, the story is a bit more complex as the force, initially repulsive, may smoothly become positive as the plates are moved apart.

The same experiment has been performed on the PVC substrate but now with three different initial volumes $V_0$ (figure 3b). In each case the force follows the same trend: it decreases following a well-defined power law, $F \propto z_1^{-3/2}$.

## 3.2. Force of a drying capillary bridge

We now fix the height $z_1$ of the capillary bridge and wait for the bridge to dry. The drying of the liquid bridge reduces slowly the volume until the bridge breaks up. As the drying is on a very large time scale, larger than thousands of seconds, the surface retraction does not create noticeable flow in the early phases of evaporation. However, some flows may exist during the drying because of the Marangoni effect. Only prior to bridge break-up, rather intense flows occur in the liquid bridge [19]. This simple experiment allows us to investigate the effect of evaporation that varies the volume without an external operation.

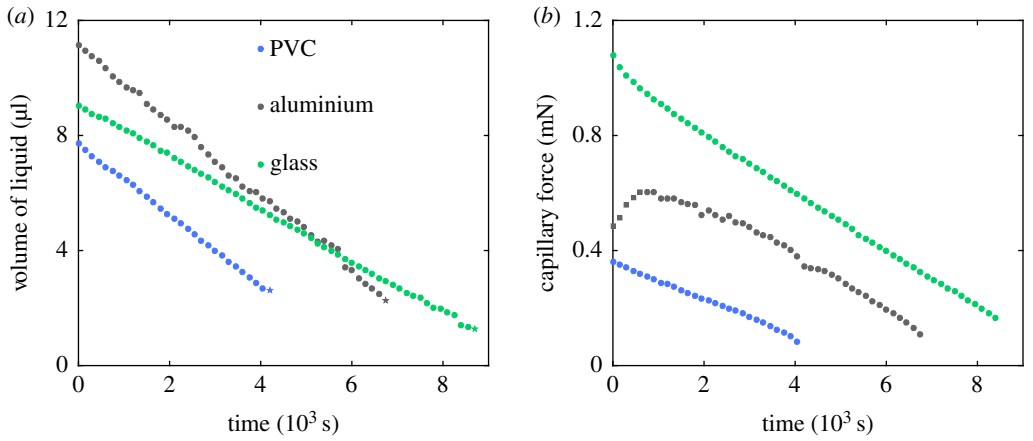

**Figure 4.** Evaporation of a liquid bridge (blue: PVC, $z_1 = 1.50$ mm; grey: aluminium, $z_1 = 1.63$ mm; green: glass, $z_1 = 1.14$ mm). (a) Direct measurement of the temporal evolution of the liquid bridge volume. The volume decreases during the drying process. (b) Direct measurement of the temporal evolution of the liquid bridge force. The force decreases during the drying process.

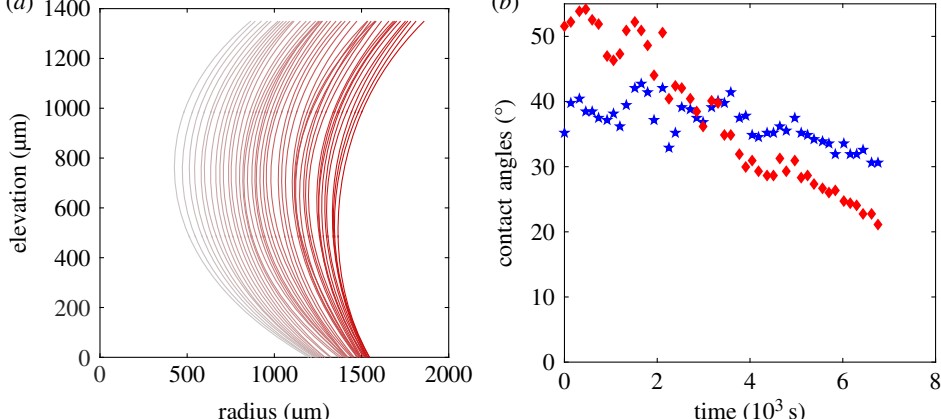

**Figure 5.** (a) One-sided profile of a drying capillary bridge on an aluminium substrate. Time evolves from dark to light. Stick–slip motion is seen at the contact line of the capillary bridge. (b) Upper (blue star) and lower (red diamond) wetting angles' temporal evolution.

The force generated by a drying capillary bridge is recorded in time as the bridge thins due to evaporation (figure 4b). The volume is also recorded temporally (figure 4a). The force, as well as the volume, decreases linearly with time except for the first time in the aluminium substrate experiment. Those outlier points are due to an effect of contact angle hysteresis. The triple line stick–slip behaviour in the early stages of the evaporation process is present both on the temporal evolution of the profile of the liquid bridge, figure 5a and in the temporal evolution of the lower wetting angle which fluctuates in the early stages of drying before decreasing more smoothly (figure 5b).

Again, in the case of aluminium, the temporal evolution of the volume is irregular because of some stick–slip behaviour of the contact line and some detection errors. The drying time (time at break-up) varies with the initial volume $V_0$ of the bridge and bridge height $z_1$.

Finally, the relationship between force and volume is reported in figure 6a. The force increases steadily and somehow linearly with the bridge volume $F \propto V$ except in the case of the aluminium for the largest volume considered. Those points correspond to the initial times where contact angle hysteresis may exist. The slope depends on the liquid height and the substrate interface properties.

## 4. Models

In this section, we rationalize the experimental results presented before (i) by developing a model of the adhesion force depending on substrate properties and geometrical parameters and (ii) by analysing the decrease of volume of the capillary bridge as a diffusion-driven evaporation. The simplicity of the

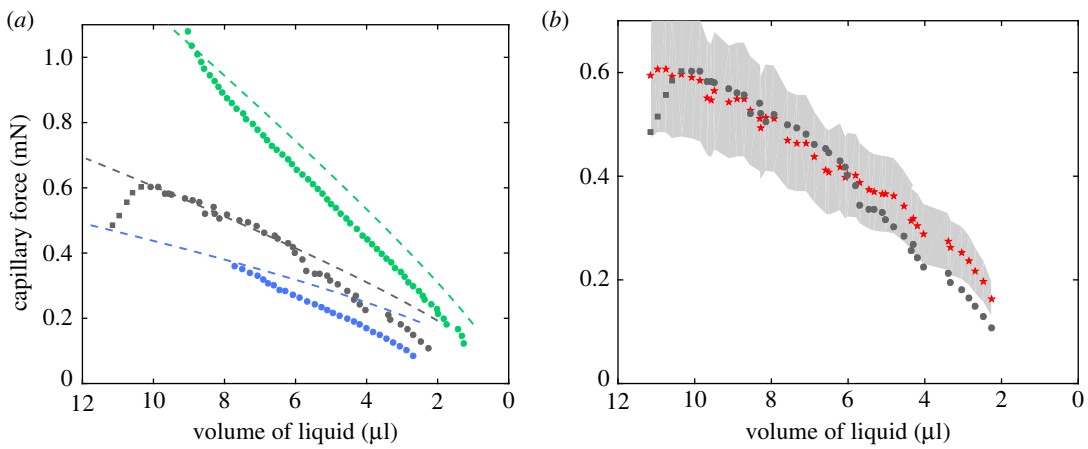

**Figure 6.** Force of a capillary bridge as a function of volume of water inside the bridge. (a) On a different substrate (blue: PVC, $z_1 = 1.50$ mm; grey: aluminium, $z_1 = 1.63$ mm; green: glass, $z_1 = 1.14$ mm). Dashed lines correspond to fits using equation (4.8) and $\theta_{PVC} = 80°$, $\theta_{glass} = 55°$ and $\theta_{aluminium} = 75°$. (b) Comparison of the force direct measurement (filled circle) and by optical measurement (star) and equation (4.7) (aluminium, $z_1 = 1.63$ mm).

geometry of the system chosen here allows one to derive analytical expressions for the two desired quantities, the force and the evaporation rate.

## 4.1. Effects of the mass transfer on the capillary bridge

The mass transfer of volatile liquid to gas creates an additional recoil pressure $P_r$ through the differential vapour recoil mechanism [20–22]. Considering the effect on mass transfer, the interface jump condition differs from the usual Young–Laplace equation by considering the recoil pressure

$$\Delta P = P_r + \gamma\mathcal{C} \quad \text{with } P_r = \left(\frac{dm}{Sdt}\right)^2\left(\frac{1}{\rho_l} - \frac{1}{\rho_g}\right), \tag{4.1}$$

where $\mathcal{C}$ is the total curvature of an interface, $dm/Sdt$ the evaporation per unit surface and $\rho_l$ and $\rho_g$ the density of the liquid and the gas.

With typical values $\gamma = 72$ mN, $V \simeq 10$ µl, $z_1 \simeq 11$ mm, $\rho_l = 1000$ kg m$^{-3}$, $\rho_g = 1$ kg m$^{-3}$ and an evaporation time $t_e \simeq 7000$ s, we can evaluate the ratio of the recoil pressure to the Laplace pressure $P_r/\gamma\mathcal{C}$. Using the following approximations, $\mathcal{C} \simeq 2/z_1 =$ and $dm/Sdt \simeq \rho_l\sqrt{V}/2\pi\sqrt{z_1}t_e$, we obtain

$$\left|\frac{P_r}{\gamma\mathcal{C}}\right| = \frac{\rho_l^2 V}{8\pi^2\rho_v t_e^2\gamma} \sim 3 \times 10^{-11} \ll 1. \tag{4.2}$$

We can thus safely neglect the effect of mass transfer on the bridge mechanics.

## 4.2. Effects of temperature on the capillary bridge

The evaporation of the liquid bridge induces temperature gradients at the liquid–air interface. Ait Saada et al. [23], for instance, showed that temperature differences at the interface of the order of 0.1°C occur during the evaporation of sessile drops of characteristic lengths 1 mm at room temperature with 40% humidity. They also showed that this result holds for conductive or insulating thick substrates. In our experiment, the effect of temperature variations is even more reduced because of two reasons: first, the evaporation flux is reduced as it occurs in a Hele-Shaw (two-dimensional flux), and experiments were performed with a humidity of 0.65, larger than 0.4 in Ait Saada et al. [23] conditions. Second, the liquid is exchanging heat on two thick substrates.

The importance of those flows can be estimated by the Marangoni Ma number [24],

$$\text{Ma} = \frac{(\partial\gamma/\partial T)\Delta Th}{\mu\alpha_T}, \tag{4.3}$$

where $\gamma$ is the surface tension, $\Delta T$ the temperature difference, $h$ the characteristic length, $\mu$ the dynamic viscosity of the liquid and $\alpha_T$ the thermal diffusivity of the liquid. For pure water, $\partial\gamma/\partial T = 1.56 \times 10^{-4}$ kg s$^{-2}$ K$^{-1}$ [25], $\Delta T \simeq 0.1$ K, $\mu = 1.0 \times 10^{-3}$ kg$^{-1}$ m$^{-1}$ s$^{-1}$ and $\alpha_T = 1.43 \times 10^{-7}$ m$^2$ s$^{-1}$

and a bridge in contact between two walls, $h = z_1/2 \simeq 0.75 \, 10^{-3}$ m, the Marangoni number is Ma $\simeq 80$. It is in the order of the critical Marangoni number above which the convective Marangoni flows occur. We thus do not expect strong Marangoni flow in our experiment. This was also pointed out by Xiao et al. [26] and Bouchenna et al. [24], who found the influence of the thermo-capillary effect on the evaporation of a sessile droplet negligible at ambient temperatures.

In the following, we will fully decouple the thermal problem from the mechanics of the capillary bridge.

## 4.3. Force of a capillary bridge

We consider here a fixed amount of liquid bridging two plates. The Bond number $B_o = \rho g z_1^2/\gamma$, where $\rho$ is the water density, compares the effect of gravity $g$ to the effect of surface tension $\gamma$. In our experiment, $B_o \sim 0.3$ means that gravitational effects, although present, could be neglected at the first order.

We will thus only take into account capillary effects in the modelling of the liquid bridge. We consider a static experiment where the fluid inside the capillary bridge is not in motion. This means that the pressure inside the liquid is constant. We model the shape of the capillary bridge in an axi-symmetric fashion with the Young–Laplace equation [27]

$$\frac{\gamma}{r(1 + \dot{r}^2)^{1/2}} - \frac{\gamma\ddot{r}}{(1 + \dot{r}^2)^{3/2}} = \Delta P, \tag{4.4}$$

where $r(z)$ is the radius of the capillary bridge at height $z$, $\dot{r} = \partial r/\partial z$ and $\Delta P$ is the pressure jump across the liquid–gas interface. This equation is accompanied by the Young–Dupré relation at the contact line that joins the three interfaces [27]

$$\left.\frac{\dot{r}}{\sqrt{1 + \dot{r}^2}}\right|_{z_1} = \cos\theta, \tag{4.5}$$

where $\theta$ is the contact angle. The Young–Dupré expression is valid if the contact line is not anchored to a defect on the solid surface. For an energetic derivation of these expressions (4.4) and (4.5), see appendix A.

The force of a capillary bridge is the sum of the effects of the normal stresses (i.e. pressure) and the triple line force. One may directly derive this expression by integration of the Young–Laplace equation (4.4), see appendix B. The force of the capillary bridge reads [28]

$$F = 2\pi\gamma r_a \cos\alpha_a \left(1 - \frac{1 - r_b \cos\alpha_b/(r_a \cos\alpha_a)}{1 - (r_b/r_a)^2}\right), \tag{4.6}$$

where $a$ and $b$ are two different elevations, $r_a = r(a)$ and $\cos\alpha_a = 1/\sqrt{1 + \dot{r}^2}\big|_a$ and, respectively, for $b$. This equation has a simple form when $(a, b)$ are taken, respectively, at the critical point (in $z_0$) and at the substrate–liquid junction in $z_1$. We obtain

$$F = 2\pi\gamma r_0 \left(1 - \frac{1 - r_1 \sin\theta/r_0}{1 - (r_1/r_0)^2}\right), \tag{4.7}$$

where $r_0 = r(z_0)$ and $r_1 = r(z_1)$. This simple expression of the force of a capillary bridge eases the measure of the force created by most of the liquid bridges by means of an optical device. In addition to the contact angle, one has only to measure two radii, the first at the contact with the substrate and the second at the critical point where the radius is minimal (hydrophilic substrate) or maximal (hydrophobic substrate). For more complex bridge shapes, where no critical point exists for instance, or when only a part of the bridge is visible, one may use the full expression of the force, equation (4.6), to compute the force generated by the liquid bridge.

The contact angle is deduced from the profile of the bridge $r(z)$ with equation (4.5) either at elevation $z = z_1$ for the upper contact angle or at elevation $z = 0$ for the lower contact angle. The contact angle and the radius at the top and at the neck are measured from the image (figure 5a,b). Those measurements are used to compute the capillary force thanks to (4.7) with air–water surface tension equal to $\gamma = 72$ mN m$^{-1}$. Those results are compared to the direct measurements of the capillary force made with the precision scale (aluminium substrate; figure 6b). As one can observe from the figure, the two measurements

follow the same trend. The deviation is of the order of the uncertainties of the optical measurements. One may note that the uncertainties of the optical measurements are much larger than the uncertainties of the measurements made with the precision scale. The limitations of such an optical measurement are threefold: (i) it is difficult to measure precisely the contact angle, (ii) the axi-symmetry may break if the triple line is anchored and (iii) the presence of gravity may affect the shape of the liquid bridge when the Bond number is larger than 0.1. Despite those limitations, the results obtained by the two approaches are in good agreement.

To rationalize the dependency of the capillary force $F$ with the height of the bridge $z_1$, we carried out a theoretical analysis by solving the Young–Laplace equation for slightly curved capillary bridges (with identical contact angles close to 90°, see appendix C). In this framework, we derived an analytical solution of the capillary force as a function of experimental parameters, bridge volume $V$, bridge height $z_1$ and contact angle $\theta$,

$$F = \gamma\sqrt{\frac{\pi V}{z_1}} + \left(\frac{2\gamma V}{z_1^2} - \frac{\pi\gamma z_1}{12}\right)\cot\theta + o(\cot^2\theta). \tag{4.8}$$

The comparison between the experimental results and the theoretical ones for the three bridge volumes shows the right trend and a pretty good agreement (figure 3b). The contact angle has been fitted to $\theta_{pvc} = 80°$ for the three cases which is larger than the observed value $\theta_{pvc} \simeq 60°$. The scaling law highlighted experimentally of the capillary force proportional to $F \propto z_1^{-3/2}$ corresponds in fact to a transition zone between two regimes where $F \propto z_1^{-2}$ and $F \propto z_1^{-1/2}$.

This expression of the capillary force, equation (4.8), has also been compared to experimental measurements of the force in figure 6. The contact angles have been fitted using $\theta_{pvc} = 80°$, $\theta_{glass} = 55°$ and $\theta_{aluminium} = 75°$, which are overestimating the real contact angles. The model gives good trend, but does not match completely the experimental data. This discrepancy is due to the assumptions considered here to solve the Young–Laplace equation which only consider slightly curved bridges and do not take into account either the stick–slip phenomenon [29] or the drift in contact angle during evaporation.

## 4.4. Drying and break-up of a capillary bridge

The liquid bridge is placed between two plates whose horizontal dimensions are much larger than the height of the liquid bridge, similar to a Hele-Shaw cell. The drying of the capillary bridge is supposed to be stationary and axi-symmetric, so that the evaporation debit $Q = -\partial V/\partial t$ is constant. The evaporation of the liquid bridge was made at room temperature between two thick plates and the slow evaporation process ensures that no strong thermal gradient existed in this experiment. In those conditions, no air convection was expected. Under this hypothesis, the water vapour is simply diffused from the liquid edge to the bulk air outside the Hele-Shaw cell $(x > L)$ following a two-dimensional Laplace problem $\partial_x(x\partial_x(\varphi))/x = 0$, where $x$ is the radial coordinate. The diffusion is driven by a gradient of water vapour concentration from a water-saturated air at the bridge surface (the partial vapour pressure equals the saturation pressure $p_v/p_v^o = 1$), to a fixed partial pressure in the bulk air outside the Hele-Shaw cell $(p_v/p_v^o = \varphi)$. Since the liquid bridge is not a regular cylinder, we will use its equivalent radius $r_e = \sqrt{V/\pi z_1}$ to describe the position of the liquid interface. Similar to the original model of Langmuir [30], the evaporation rate reads as

$$\frac{V - V_0}{z_1(1 - \varphi)} = \frac{2\pi D M_w P_v^o}{\rho RT \ln(L/r_e)}t, \tag{4.9}$$

where $D$ is the diffusion coefficient of water vapour in air, $M_w$ the molar mass of water, $R$ the perfect gas constant and $T$ the temperature. This relation is obtained by integration assuming that the logarithm does not change much during the drying process. The evaporation debit is simply proportional to the height of the bridge $z_1$, but depends logarithmically on the dimension of the Hele-Shaw cell $L$. The experiments were carried out with relative humidity $\varphi$ of $0.65 \pm 0.05$. All the experiments collapse on a master curve when drawing the quantity $(V(t) - V_0)/z_1(1 - \varphi)$ (figure 7). This collapse shows that the evaporation rate is constant all through the process until break-up and proportional to the height of the liquid bridge. Finally, we compare the prediction, equation (4.9), to the experimental data with a reasonable agreement. The discrepancies come from the uncertainties on the relative humidity $\varphi$ and on the volume measurement.

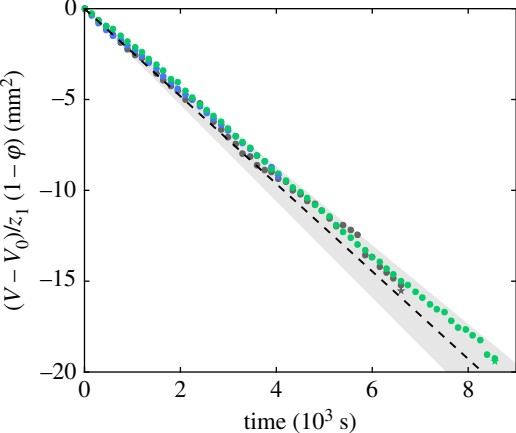

**Figure 7.** Diffusion-controlled evaporation of a liquid bridge. Direct comparison of the experimental evaporation debit with the diffusion-controlled evaporation model, equation (4.9).

In a more accurate integration of the evaporation process, evaporation rate changes the radius $r_e$ by the relation $Q = -\partial V/\partial t = -2\pi z_1 r_e \partial r_e/\partial t$. One may get through the integration of equation (4.9) the relation between $r_e$ and $t$ as

$$\left(\frac{\pi r_e^2 z_1}{2V_0}\right)\left[1 - \ln\left(\frac{r_e^2}{L^2}\right)\right] = \frac{1}{2}\left[1 - \ln\left(\frac{V_0}{\pi z_1 L^2}\right)\right] - \frac{2\pi z_1 DM_w P_v^o(1-\varphi)}{V_0 \rho RT}t, \tag{4.10}$$

where $r_i$ is the initial equivalent radius. The rupture of the capillary bridge occurs when $r_e \to 0$. This simply leads to the typical evaporation time $t_e$,

$$t_e = \frac{V_0 \rho RT}{4\pi z_1 DM_w P_v^o(1-\varphi)}\left[1 - \ln\left(\frac{V_0}{\pi z_1 L^2}\right)\right]. \tag{4.11}$$

The typical evaporation time is slightly larger than the break-up time, since at break-up, the equivalent radius is not exactly null. It is worth noting that the break-up may depend on the nature of the substrate on which the liquid bridge has been deposited [3]. Indeed, the break-up occurs when the bridge cannot meet both Young–Laplace and Young–Dupré equations (contact angle) for a given volume. We then do expect that the equivalent radius at break-up may slightly depend on the substrate nature through the contact angle.

# 5. Conclusion and perspectives

We have built a set-up to study both surface and evaporation properties of liquids. This set-up allows one to measure several time-dependent properties of a drying liquid bridge, such as volume, surface and force. The adhesive force generated by a liquid bridge has been observed to vary proportionally with the liquid bridge volume $F \propto V$ and inversely with the height of the bridge according to equation (4.8). The ageing of contact angles must be taken into account for more precise modelling. The force generated by a liquid bridge may be fairly estimated by optical measurement of the liquid bridge shape and equation (4.7). The evaporation of the liquid bridge is mainly diffusion-driven when it is confined in two dimensions; convection in the vicinity of the capillary bridge is negligible due to the confinement. The two support plates act as a Hele-Shaw cell in which water vapour only diffuses. This set-up well defines the evaporation framework. We showed from this analysis that the diffusion-driven evaporation has no influence on the capillary forces.

This work opens two research areas: (i) the first is to model the dependency of the force of a capillary bridge on the liquid height, liquid volume, surface tension and contact angles as easily usable quantities; (ii) the second is to use the set-up for analysing active fluids. The surface properties of active fluids are a key element to understand the spreading and the survival of microbes. This standard set-up allows one to characterize the evolution in time of the surface tension and the evaporation rate of a active-liquid bridge. We aim at applying our automated set-up to characterize the effects of bio-films and bio-secretions to the interface properties of fluids.

Data accessibility. All data are made available in the electronic supplementary material.
Authors' contributions. O.R. and Lounès Tadrist designed the experiment. L.M. performed data acquisition. Loïc Tadrist and L.M. performed data treatment. Loïc Tadrist made the modelling. Loïc Tadrist and Lounès Tadrist wrote the paper.
Competing interests. The authors declare not having any competing interest.
Funding. This work was partially supported by the FNRS (grant no. CHAR.RECH.-1.B423.18, Loïc Tadrist).
Acknowledgements. Authors acknowledge fruitful discussions with Tristan Gilet, Youness Tourtit, Antonio Iazzolino, Pierre Lambert, Salah Chikh and Mebrouk Ait Saada.

# Appendix A. Description of an axi-symmetric capillary bridge

The Young–Dupré and Young–Laplace equations may be computed from an energy minimization problem with a fixed liquid volume. If the triple line is not anchored to any defects of the substrate, the liquid bridge can be described as a cylinder defined by its radius $r(z)$. The surface $S$ and the volume $V$ of the liquid bridge in the coordinates given in the figure 2 read

$$S[r] = \int_0^{z_1} 2\pi r \sqrt{1 + \dot{r}^2}\, dz \quad \text{and} \quad V[r] = \int_0^{z_1} \pi r^2\, dz. \tag{A 1}$$

For the sake of simplicity, we consider a case where gravity does not play any role, corresponding to small Bond numbers $B_o \to 0$. In those conditions, if the upper and lower substrates are identical, the capillary bridge should be symmetric $r(z + z_1/2) = r(-z + z_1/2)$. The surface energy of the bridge is the functional $E[r] = \gamma S[r] + 2\pi(\gamma_{SL} - \gamma_{SG})\, r_1^2$, where $\gamma$ is the surface energy of the liquid–gas interface, $\gamma_{SL}$ and $\gamma_{SG}$, and $r_1 = r(z_1)$ is the radius of the liquid bridge in contact with the solid. The surface energy of the bridge is simply the energy of the liquid–gas interface plus the energy of the liquid–solid interface diminished by the energy of the solid–gas interface replaced by the liquid–solid interface. In order to apply the Lagrange multiplier method, we add the constant $\lambda V$ to the functional $E[r]$ where $\lambda$ is the Lagrange multiplier. This will have no influence on the results since energy is defined through a constant. This trick allows one to fix directly the volume in the expressions. Finally, we aim at minimizing the functional

$$E[r] = \int_0^{z_1} 2\pi\gamma r \sqrt{1 + \dot{r}^2} - \lambda\pi r^2 \ dz + 2\pi(\gamma_{SL} - \gamma_{SG})\, r_1^2. \tag{A 2}$$

We now vary $E[r]$ around $r(z)$ by a small function around $\epsilon\delta_r(z)$, where $\epsilon$ is the amplitude of the variation.

$$E[r + \epsilon\delta_r] = \pi \int_0^{z_1} 2\gamma(r + \epsilon\delta_r)\sqrt{1 + (r + \dot{\epsilon\delta_r})^2} - \lambda(r + \epsilon\delta_r)^2 \ dz + 2\pi(\gamma_{SL} - \gamma_{SG})(r + \epsilon\delta_r)^2$$

$$+ \pi(\gamma_{SL} - \gamma_{SG})\,(r + \delta r)_{z_1}^2$$

$$= E[r] + \pi\epsilon \underbrace{\left( \int_0^{z_1} A(x)\, dz + 2(\gamma_{SL} - \gamma_{SG})r_1\delta_{r_1} \right)}_{\text{First-order term: } G} + \mathcal{O}(\epsilon^2).$$

At this point, we have the first-order variation as

$$G = \int_0^{z_1} 2\gamma\sqrt{1 + \dot{r}^2}\delta_r + \frac{2\gamma r\dot{r}}{\sqrt{1 + \dot{r}^2}}\dot{\delta}_r - 2\lambda r\delta_r\ dz + 2(\gamma_{SL} - \gamma_{SG})r_1\delta_{r_1}$$

$$= \left[ \frac{2\gamma r\dot{r}}{\sqrt{1 + \dot{r}^2}}\delta_r \right]_0^{z_1} + \int_0^{z_1} \left\{ 2\gamma\sqrt{1 + \dot{r}^2}\delta_r - \frac{2\gamma\dot{r}^2}{\sqrt{1 + \dot{r}^2}} - \frac{2\gamma r\ddot{r}}{\sqrt{1 + \dot{r}^2}} \right.$$

$$\left. + \frac{2\gamma r\dot{r}^2\ddot{r}}{(1 + \dot{r}^2)^{3/2}} - 2\lambda r \right\} \delta_r\ dz + 2(\gamma_{SL} - \gamma_{SG})r_1\delta_{r_1}$$

$$= 2\int_0^{z_1} \left\{ \frac{\gamma}{(1 + \dot{r}^2)^{1/2}} - \frac{\gamma r\ddot{r}}{(1 + \dot{r}^2)^{3/2}} - \lambda r \right\} \delta_r\ dz + 2 \left( (\gamma_{SL} - \gamma_{SG}) + \gamma\frac{\dot{r}}{\sqrt{1 + \dot{r}^2}}\bigg|_{z_1} \right) r_1\delta_{r_1}.$$

Extrema of functional $E$ are found when for any $\delta r$,

$$\pi G = \frac{E[r + \epsilon\delta_r] - E[r]}{\epsilon}\underset{\epsilon\to 0}{=} 0. \tag{A 3}$$

This is realized when

$$\underbrace{\frac{\gamma}{(1+\dot{r}^2)^{1/2}} - \frac{\gamma r \ddot{r}}{(1+\dot{r}^2)^{3/2}} - \lambda r = 0}_{\text{Young–Laplace}} \quad \text{and} \quad \underbrace{\gamma_{\text{SL}} - \gamma_{\text{SG}} + \gamma \frac{\dot{r}}{\sqrt{1+\dot{r}^2}}\bigg|_{z_1} = 0.}_{\text{Young–Dupré}} \tag{A 4}$$

The first equation is also known as the Young–Laplace equation where $\lambda$ is interpreted as the pressure jump $\Delta P$ through the interface. The second equation is simply the Young–Dupré equation where you just remark that $\cos\theta = (\dot{r}/\sqrt{1+\dot{r}^2})_{z_1}$. We have considered here a symmetric capillary bridge. One would have studied with almost the same calculation, a non-symmetric liquid bridge where the upper substrate and the lower substrate are different. This would have brought two Young–Dupré equations instead of only one.

We proved that the Young–Laplace equation corresponds to an extremum of the functional $E[r]$. This extremum goes from a stable equilibrium to an unstable one which will cause the liquid bridge to break up when the liquid bridge gets stretched.

# Appendix B. Normal force of a liquid bridge

In the quasi-static regime, the capillary force $F$ is constant along the liquid bridge. The force is simply the indefinite integral of the Young–Laplace equation (4.4). One finds

$$F = -\pi r^2 \Delta P + 2\pi\gamma \frac{r}{\sqrt{1+\dot{r}^2}}. \tag{B 1}$$

This force has an extremely simple interpretation, the first part being the action of the pressure forces and the second part being the effect of the line tension. Those two forces, the pressure force and the line tension force, both originate from the surface phenomena. One may note that the line tension part of the liquid bridge force, $2\pi\gamma r/\sqrt{1+\dot{r}^2}$, is always positive and always tends to gather the two parts of the plates. Less obvious are the pressure forces that can be either positive or negative depending on the sign of the mean curvature of the surface. Quite surprisingly, just before break-up, the pressure inside the liquid is larger than in the surrounding air: the pressure forces tend to mitigate the effect of the line tension forces.

The force of the capillary bridge, as well as the pressure inside the liquid, are two constants that do not depend on the elevation $z$. One may evaluate the expression (B 1) at two different elevations, let us say $a$ and $b$,

$$\left.\begin{array}{l} F = -\pi r_a^2 \Delta P + 2\pi\gamma r_a/\sqrt{1+\dot{r}_a^2} \\[2mm] F = -\pi r_b^2 \Delta P + 2\pi\gamma r_b/\sqrt{1+\dot{r}_b^2} \end{array}\right\} \tag{B 2}$$

and

To simplify the expressions, we will use in the following the angle $\alpha(z)$ defined in figure 2, with $\cos\alpha = 1/\sqrt{1+\dot{r}^2}$. The linear system of equations (B 2) can be solved to obtain both the expression of the capillary force of the liquid bridge as well as the pressure jump across the surface, they read

$$\Delta P = 2\gamma\left(\frac{r_a \cos\alpha_a - r_b \cos\alpha_b}{r_a^2 - r_b^2}\right) \quad \text{and} \quad F = 2\pi\gamma r_a \cos\alpha_a\left(1 - \frac{1 - r_b \cos\alpha_b/r_a \cos\alpha_a}{1 - r_b^2/r_a^2}\right). \tag{B 3}$$

This expression might be used to measure with optical means the pressure and the force of any axi-symmetric capillary bridge. However, from an experimental point of view, it requires to obtain a very accurate measurement of the radius since its derivative is involved to compute the angle $\alpha$. An even more useful expression of the force might be used through considering $b = z_1$ being the contact point between the liquid bridge and the substrate and $a = z_0$ being the critical point where $\dot{r} = 0$. At these locations, only the radii are required to obtain the capillary force

$$F = 2\pi\gamma r_0\left(1 - \frac{1 - r_1 \sin\theta/r_0}{1 - (r_1/r_0)^2}\right). \tag{B 4}$$

# Appendix C. Solution for slightly curved axi-symmetric bridges

For slightly curved bridges, we can restrict the solution to the Young–Laplace equation at the second order, $r(x) = r(x) + o(x^3)$ with $r(x) = a_0 + a_1\, x + a_2\, x^2$. The coefficients $a_1$ and $a_2$ have simple expressions as a function of the neck radius $a_0$, the force $F$ and the pressure jump $\Delta P$

$$a_1 = \frac{\sqrt{(2\pi\gamma)^2 a_0 h - (F + \pi\Delta P a_0^2)^2}}{F + \pi\Delta P a_0^2} \quad \text{and} \quad a_2 = 2(\pi\gamma)^2 a_0 \frac{F - \pi\Delta P a_0^2}{(F + \pi\Delta P a_0^2)^3}. \tag{C 1}$$

This choice of coefficients solve equation (B 1) in series at the order 3. With the limit condition $\dot{r}(h/2) = \cot\theta$, the symmetry condition $r(x) = r(-x)$ and the volume conservation $V = \pi \int_{-h/2}^{h/2} r^2 \mathrm{d}x$, we find the expression of the force and pressure inside the bridge

$$\frac{F}{2\pi\gamma h} = \frac{1}{2}\sqrt{\frac{V}{h^3\pi} - \frac{\cot^2\theta}{180}} - \frac{\cot\theta}{24} + \cot\theta\left(\frac{V}{h^3\pi} - \frac{\cot\theta}{6}\sqrt{\frac{V}{\pi} - \frac{\cot^2\theta}{180}} + \frac{1}{720}\cot^2\theta\right) \tag{C 2}$$

and

$$\frac{\Delta P h}{2\gamma} = \left(2\sqrt{\frac{V}{h^3\pi} - \frac{\cot^2\theta}{180}} - \frac{\cot\theta}{6}\right)^{-1} - \cot\theta, \tag{C 3}$$

which may be simplified in the limit of slightly curved bridges, $|\cot\theta| \ll 1$,

$$F = \gamma\sqrt{\frac{\pi V}{h}} + \left(\frac{2\gamma V}{h^2} - \frac{\pi\gamma h}{12}\right)\cot\theta + o(\cot^2\theta) \tag{C 4}$$

and

$$\Delta P = \gamma\sqrt{\frac{\pi h}{V}} + \gamma\cot\theta\left(\frac{\pi h^2}{12V} - \frac{1}{h}\right) + o(\cot^2\theta). \tag{C 5}$$

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
