## [Reviewer comments · Royal Society Open Science]

Review History

RSOS-182016.R0 (Original submission)

Review form: Reviewer 1

Is the manuscript scientifically sound in its present form?

No

Are the interpretations and conclusions justified by the results?

No

Is the language acceptable?

Yes

Is it clear how to access all supporting data?

Yes

Do you have any ethical concerns with this paper?

No

Have you any concerns about statistical analyses in this paper?

I do not feel qualified to assess the statistics

Recommendation?

Major revision is needed (please make suggestions in comments)

Comments to the Author(s)

The present paper presents experimental results on the dynamic evolution of a capillary bridge, trapped between two plates and subjected to evaporation.

This configuration is claimed to be relevant for different biochemical or biotechnological applications.

A bit of literature on capillary bridges is mentioned albeit very quickly since the configuration here studied, as well as the associated objectives and experiments, have also been extensively studied by a research group of Limoges Univ., both in hydrophilic and hydrophobic cases. This part of existing literature is passed over in silence in the present paper (see doi: 10.4236/amc.2016.67017 and doi: 10.4236/amc.2017.74009, *Advances in Materials Physics and Chemistry*).

As argued by the authors in the end, a capillary bridge is considered as relevant to investigating the effects of bio-films on the interfacial properties. But in fact, this is all the more debatable given that the literature on the evaporation of sessile drops is much more advanced and that sessile drops are classically used in bio-microfluidics. For instance, different effects due to thermal Marangoni effect and local enhancement of evaporation at contact lines are already known and systematically studied at drop scale.

One main question remains after reading the article: why should one study a capillary bridge rather than a sessile drop if the objective is to study the behaviour of an interface under the effect of evaporation?

Q0. What are the distinguishing arguments?

Regarding the content of the article, many points must be investigated much more in details.

Q1. Evaporation kinetics with glass and PVC plates is found different at short time from that with aluminum plates (Fig. 4b). Surface state is briefly evoked as a possible reason for this. It must be checked by testing differentially polished aluminum supports.

Q2. It can be highly recommended to measure the temperature distribution along the two plates so as to check that whatever the nature of the materials is (thermally conductive or not), the whole configuration can be considered as only driven by a small Biot number. A natural question arises as to whether the boundary conditions for the thermal problem are really considered by the authors. No mention to this particular point is made in the paper since only the room temperature is given.

C1. Fig. 5a is just a combination of two previous figures with no decisive input.

Q4. The authors do not provide time-dependent profiles of the capillary bridge, $r(t,z)$, despite the possibility to get them from imaging.

Why?

These profiles reveal to be a decisive information, especially if the aim is to give an idea of the dynamic behaviour of the wetting angles and the triple contact line along the plates: stick-slip, no-slip motion...?

Q5. The Young-Laplace and Young-Dupré equations are derived, based on an energetic point of view, in order to calculate the normal force between the two plates and to justify the boundary condition based on wetting angles. In fact, this model is nothing but a simplified form of a normal momentum balance at the capillary bridge surface ("Interfacial Transport Phenomena", Springer, by Slattery, Sagis & Oh).

What are the assumptions which make this mechanical model decoupled from the diffusion-limited (evaporation) model? What is the value of the Bond number which allows for neglecting gravity effects?

C2. Figure 6. The caption can not be related to the data points.

Q6. The diffusion-limited model for the evaporation as proposed in page 7 is a clone of the simple model proposed by Bexon & Picknett (The evaporation of sessile or pendant drops in still air, Journal of Colloid and Interface Science 61 (2) (1977) 336–350).

The additional introduction of an equivalent radius r_e all along the capillary bridge is not necessary and a full expression based on a local time-dependent radius must be implemented. Of course, this makes the model non-linear but nowadays, this is no more an issue.

Q7. Evaporation is well-known to focus at the contact lines of sessile drops, yielding a thermally-induced Marangoni flow. What happens here with a capillary bridge? Can the authors justify why they do not consider thermally-induced Marangoni effect?

Q8. The role of convection at the vicinity of the capillary bridge is not mentioned. Is it negligible? If so, why?

Taking into account all the elements mentioned above, I think that the paper is far from ready for publication. It must distinguish itself (if possible) from the existing literature, especially by taking into account the experimental results already published.

The modelling approach is oversimplified, which is not a basic issue provided that this is extensively justified in terms of dimensionless numbers on the basis of a generalized model.

For now, this is not the case.

Review form: Reviewer 2

Is the manuscript scientifically sound in its present form?

No

Are the interpretations and conclusions justified by the results?

Yes

Is the language acceptable?

Yes

Is it clear how to access all supporting data?

Yes

Do you have any ethical concerns with this paper?

No

Have you any concerns about statistical analyses in this paper?

No

Recommendation?

Accept with minor revision (please list in comments)

Comments to the Author(s)

The manuscripts presents evolution of capillary bridges between two large parallel plates. Evolution of capillary force and mass loss due to evaporation and extension is assessed experimentally. Capillary force is also calculated using Laplace equation, based on the measured geometrical characteristics of liquid bridge: gorge radius, contact radius and contact angle. Evaporation rate is calculated with the use of mass balance. Comparison of intergranular force measurement from the experiments and calculated from Laplace-Young law is presented. They suggest to use evaporation dynamics of a liquid bridge between two parallel plates of different solids to determine surface characteristics.

The paper is relatively well structured and the concept is well explained. Should be published after minor corrections. Minor shortcomings are indicated below.

P1

The title is awkward (Capillary bridge evaporation study: a robust method to characterize the interface properties of fluids)

- L. 25: It should be "key element" (not key bock)

- L. 29: The whole Abstract should be more informative, and specific. English should be revised

P4

- 52: "no noticeable flow occurs in the bridge and it can be considered at rest" – This statement needs to be qualified. There are findings available for other evaporating capillary bridges, sessile drops indicating strong pressure gradients and flows within the bridge as well, especially intense shortly prior bridge rupture (see e.g. Snoeijer and Anreotti, 2013, Yang et al., 2019). Coffee stain flows are the best known examples of such flows. Here, the safest would be to add that "in the early phases of evaporation".

P5

Fig. 5- would be more intuitive to use the horizontal axis along the process progress. Invert the horizontal axis, from highest to lowest volume.

P6

- 31: Bond number (not bond number)

P7

- 18- how contact angle was measured? In which distance from contact point? Indicate what does it mean upper or lower contact angle? Calculated capillary force is very sensitive to contact angle changes

- 33: Should be: bridge was placed between two plates ? L 50-56

P8

Figure 6: where are a) b) and c)?

P10

- 56- 58- The sentence is not clear.

Decision letter (RSOS-182016.R0)

15-Mar-2019

Dear Dr Tadrst:

Manuscript ID RSOS-182016 entitled "Evaporation of a capillary of a bridge: a robust method to characterize the interface properties of fluids." which you submitted to Royal Society Open

Science, has been reviewed. The comments from reviewers are included at the bottom of this letter.

In view of the criticisms of the reviewers, the manuscript has been rejected in its current form. However, a new manuscript may be submitted which takes into consideration these comments.

Please note that resubmitting your manuscript does not guarantee eventual acceptance, and that your resubmission will be subject to peer review before a decision is made.

Your resubmitted manuscript should be submitted by 12-Sep-2019. If you are unable to submit by this date please contact the Editorial Office.

on behalf of Professor Alban Potherat (Associate Editor) and Professor R. Kerry Rowe (Subject Editor)
openscience@royalsociety.org

Associate Editor Comments to Author (Professor Alban Potherat):

While one of the referees is rather positive, the other referee raises very serious concerns about the manuscript. In particular, this referee feels that the originality and importance of the scientific contribution is not demonstrated against existing literature and that the scientific assumptions are not sufficiently justified to provide confidence in the results.

The concerns take the manuscript beyond the scope of a major revision, but the the work can still provide the basis for a later re submission, provided all comments from the referees are carefully addressed.

Reviewers' Comments to Author:

Reviewer: 1

Comments to the Author(s)

The present paper presents experimental results on the dynamic evolution of a capillary bridge, trapped between two plates and subjected to evaporation.

This configuration is claimed to be relevant for different biochemical or biotechnological applications.

A bit of literature on capillary bridges is mentioned albeit very quickly since the configuration here studied, as well as the associated objectives and experiments, have also been extensively

studied by a research group of Limoges Univ., both in hydrophilic and hydrophobic cases. This part of existing literature is passed over in silence in the present paper (see doi: 10.4236/amcp.2016.67017 and doi: 10.4236/amcp.2017.74009, *Advances in Materials Physics and Chemistry*).

As argued by the authors in the end, a capillary bridge is considered as relevant to investigating the effects of bio-films on the interfacial properties. But in fact, this is all the more debatable given that the literature on the evaporation of sessile drops is much more advanced and that sessile drops are classically used in bio-microfluidics. For instance, different effects due to thermal Marangoni effect and local enhancement of evaporation at contact lines are already known and systematically studied at drop scale.

One main question remains after reading the article: why should one study a capillary bridge rather than a sessile drop if the objective is to study the behaviour of an interface under the effect of evaporation?

Q0. What are the distinguishing arguments?

Regarding the content of the article, many points must be investigated much more in details.

Q1. Evaporation kinetics with glass and PVC plates is found different at short time from that with aluminum plates (Fig. 4b). Surface state is briefly evoked as a possible reason for this. It must be checked by testing differentially polished aluminum supports.

Q2. It can be highly recommended to measure the temperature distribution along the two plates so as to check that whatever the nature of the materials is (thermally conductive or not), the whole configuration can be considered as only driven by a small? Biot number. A natural question arises as to whether the boundary conditions for the thermal problem are really considered by the authors. No mention to this particular point is made in the paper since only the room temperature is given.

C1. Fig. 5a is just a combination of two previous figures with no decisive input.

Q4. The authors do not provide time-dependent profiles of the capillary bridge, $r(t,z)$, despite the possibility to get them from imaging.

Why?

These profiles reveal to be a decisive information, especially if the aim is to give an idea of the dynamic behaviour of the wetting angles and the triple contact line along the plates: stick-slip, no-slip motion...?

Q5. The Young-Laplace and Young-Dupré equations are derived, based on an energetic point of view, in order to calculate the normal force between the two plates and to justify the boundary condition based on wetting angles. In fact, this model is nothing but a simplified form of a normal momentum balance at the capillary bridge surface ("*Interfacial Transport Phenomena*", Springer, by Slattery, Sagis & Oh).

What are the assumptions which make this mechanical model decoupled from the diffusion-limited (evaporation) model? What is the value of the Bond number which allows for neglecting gravity effects?

C2. Figure 6. The caption can not be related to the data points.

Q6. The diffusion-limited model for the evaporation as proposed in page 7 is a clone of the simple model proposed by Bexon & Picknett (*The evaporation of sessile or pendant drops in still air*, *Journal of Colloid and Interface Science* 61 (2) (1977) 336–350).

The additional introduction of an equivalent radius r_e all along the capillary bridge is not necessary and a full expression based on a local time-dependent radius must be implemented. Of course, this makes the model non-linear but nowadays, this is no more an issue.

Q7. Evaporation is well-known to focus at the contact lines of sessile drops, yielding a thermally-induced Marangoni flow. What happens here with a capillary bridge? Can the authors justify why they do not consider thermally-induced Marangoni effect?

Q8. The role of convection at the vicinity of the capillary bridge is not mentioned. Is it negligible? If so, why?

Taking into account all the elements mentioned above, I think that the paper is far from ready for publication. It must distinguish itself (if possible) from the existing literature, especially by taking into account the experimental results already published.

The modelling approach is oversimplified, which is not a basic issue provided that this is extensively justified in terms of dimensionless numbers on the basis of a generalized model. For now, this is not the case.

Reviewer: 2

Comments to the Author(s)

The manuscript presents evolution of capillary bridges between two large parallel plates. Evolution of capillary force and mass loss due to evaporation and extension is assessed experimentally. Capillary force is also calculated using Laplace equation, based on the measured geometrical characteristics of liquid bridge: gorge radius, contact radius and contact angle. Evaporation rate is calculated with the use of mass balance. Comparison of intergranular force measurement from the experiments and calculated from Laplace-Young law is presented. They suggest to use evaporation dynamics of a liquid bridge between two parallel plates of different solids to determine surface characteristics.

The paper is relatively well structured and the concept is well explained. Should be published after minor corrections. Minor shortcomings are indicated below.

P1

The title is awkward (Capillary bridge evaporation study: a robust method to characterize the interface properties of fluids)

- L. 25: It should be "key element" (not key bock)

- L. 29: The whole Abstract should be more informative, and specific. English should be revised

P4

- 52: "no noticeable flow occurs in the bridge and it can be considered at rest" - This statement needs to be qualified. There are findings available for other evaporating capillary bridges, sessile drops indicating strong pressure gradients and flows within the bridge as well, especially intense shortly prior bridge rupture (see e.g. Snoeijer and Anreotti, 2013, Yang et al., 2019). Coffee stain flows are the best known examples of such flows. Here, the safest would be to add that "in the early phases of evaporation".

P5

Fig. 5- would be more intuitive to use the horizontal axis along the process progress. Invert the horizontal axis, from highest to lowest volume.

P6

- 31: Bond number (not bond number)

P7

- 18- how contact angle was measured? In which distance from contact point? Indicate what does it mean upper or lower contact angle? Calculated capillary force is very sensitive to contact angle changes

- 33: Should be: bridge was placed between two plates ? L 50-56
P8
- Figure 6: where are a) b) and c)?
P10
- 56- 58- The sentence is not clear.

Author's Response to Decision Letter for (RSOS-182016.R0)

See Appendices A & B.

RSOS-190887.R0

Review form: Reviewer 1

Is the manuscript scientifically sound in its present form?

No

Are the interpretations and conclusions justified by the results?

Yes

Is the language acceptable?

Yes

Is it clear how to access all supporting data?

Not Applicable

Do you have any ethical concerns with this paper?

No

Have you any concerns about statistical analyses in this paper?

I do not feel qualified to assess the statistics

Recommendation?

Major revision is needed (please make suggestions in comments)

Comments to the Author(s)

I am a little bit surprised to see a revised version after the paper has been rejected.

I can see that the papers by Portuguez et al have been cited but the data have not been taken into account for comparison purposes.

Considering now the modelling part of the paper, I think that the authors would gain much benefit from considering the paper by Chen et al. (<https://doi.org/10.1021/la304870h>), which is not cited here!

My question Q5 (first review) on the normal momentum balance at the liquid surface is not properly addressed. Mass transfers are commonly responsible for an additional force whose (negligible?) role must be properly made evident.

Considering the previous points, it is difficult to recommend publication of the paper in its present form.

Review form: Reviewer 3 (Boryan Radoev)

Is the manuscript scientifically sound in its present form?

Yes

Are the interpretations and conclusions justified by the results?

No

Is the language acceptable?

Yes

Is it clear how to access all supporting data?

Yes

Do you have any ethical concerns with this paper?

No

Have you any concerns about statistical analyses in this paper?

No

Recommendation?

Accept with minor revision (please list in comments)

Comments to the Author(s)

I found very interesting the experimentally established dependencies, $F \sim z^{-3/2}$ at constant volume V (Fig.4) and $F \sim 1/V$ at constant height, z (Fig.6), but they need a theoretical analysis. Taking into account that in the literature can be found analytical solution of the bridge height z and volume V as function of the same parameters (r_0, r_1, θ) as the parameters of the force (see e.g. Capillary Bridges in Wikipedia and the literature cited there) such an analysis is only a routine procedure. Moreover, graph in Figure 6b looks more as a curve (with negative second derivative) than as a straight line. Do that analysis and the study will be a very useful and instructive paper.

Decision letter (RSOS-190887.R0)

10-Jul-2019

Dear Dr Tadrict:

Manuscript ID RSOS-190887 entitled "Characterisation of interface properties of fluids by evaporation of a capillary bridge." which you submitted to Royal Society Open Science, has been reviewed. The comments from reviewer(s) are included at the bottom of this letter.

In view of the criticisms of the reviewer(s), I must decline the manuscript for publication in Royal Society Open Science at this time. However, a new manuscript may be submitted which takes into consideration these comments.

Please note that resubmitting your manuscript does not guarantee eventual acceptance, and that your resubmission will be subject to re-review by the reviewer(s) before a decision is rendered.

You will be unable to make your revisions on the originally submitted version of your manuscript. Instead, revise your manuscript using a word processing program and save it on your computer.

You may also click the below link to start the resubmission process (or continue the process if you have already started your resubmission) for your manuscript. If you use the below link you will not be required to login to ScholarOne Manuscripts.

*** PLEASE NOTE: This is a two-step process. After clicking on the link, you will be directed to a webpage to confirm. ***

https://mc.manuscriptcentral.com/rsos?URL_MASK=ffb2a6abe5984ec1b740d605a57de65e

Because we are trying to facilitate timely publication of manuscripts submitted to Royal Society Open Science, your resubmitted manuscript should be submitted by 07-Jan-2020. If you are unable to submit by this date please contact the Editorial Office for options.

I look forward to a resubmission.

on behalf of Professor Alban Potherat (Associate Editor) and R. Kerry Rowe (Subject Editor)
openscience@royalsociety.org

Associate Editor Comments to Author (Professor Alban Potherat):

Both referees indicate that the theoretical model is insufficient to provide a satisfactory explanation for the experimental results. This was already pointed out at the first review but the referees are not satisfied that the issue has been properly addressed. As such, the manuscript does not bring a contribution that would justify of a publication. Should you decide to resubmit your manuscript at a later stage, these points would have to be convincingly addressed.

Reviewer comments to Author:

Reviewer: 1

Comments to the Author(s)

I am a little bit surprised to see a revised version after the paper has been rejected.

I can see that the papers by Portuguez et al have been cited but the data have not been taken into account for comparison purposes.

Considering now the modelling part of the paper, I think that the authors would gain much benefit from considering the paper by Chen et al. (<https://doi.org/10.1021/la304870h>), which is not cited here!

My question Q5 (first review) on the normal momentum balance at the liquid surface is not properly addressed. Mass transfers are commonly responsible for an additional force whose (negligible?) role must be properly made evident.

Considering the previous points, it is difficult to recommend publication of the paper in its present form.

Reviewer: 3

Comments to the Author(s)

I found very interesting the experimentally established dependencies, $F \sim z^{-3/2}$ at constant volume V (Fig.4) and $F \sim 1/V$ at constant height, z (Fig.6), but they need a theoretical analysis. Taking into account that in the literature can be found analytical solution of the bridge height z and volume V as function of the same parameters (r_0 , r_1 , θ) as the parameters of the force (see e.g. Capillary Bridges in Wikipedia and the literature cited there) such an analysis is only a routine procedure. Moreover, graph in Figure 6b looks more as a curve (with negative second derivative) than as a straight line. Do that analysis and the study will be a very useful and instructive paper.

Author's Response to Decision Letter for (RSOS-190887.R0)

See Appendices C & D.

RSOS-191608.R0

Review form: Reviewer 1

Is the manuscript scientifically sound in its present form?

Yes

Are the interpretations and conclusions justified by the results?

Yes

Is the language acceptable?

Yes

Do you have any ethical concerns with this paper?

No

Have you any concerns about statistical analyses in this paper?

No

Recommendation?

Accept with minor revision (please list in comments)

Comments to the Author(s)

I am rather satisfied with this last response though it could be actually interesting to investigate much more the case of a metallic substrate whose role on evaporation kinetics was not really clarified. Ageing of the dynamic contact angle stands also as a decisive ingredient to be taken into account for any future modeling.

These two points must be clearly mentioned in the conclusion.

Review form: Reviewer 3 (Boryan Radoev)

Is the manuscript scientifically sound in its present form?

Yes

Are the interpretations and conclusions justified by the results?

Yes

Is the language acceptable?

Yes

Do you have any ethical concerns with this paper?

No

Have you any concerns about statistical analyses in this paper?

No

Recommendation?

Accept with minor revision (please list in comments)

Comments to the Author(s)

“Characterization of interface properties of fluids by evaporation of a capillary bridge” by L. Tadriss¹, L. Motte², O. Rahli² and L.

Here are my comments to Appendix C and to the answer of my review.

In general, Appendix C meets the requirement of an analytical derivation of Eq. (4.8), but at a carefully reading one finds still some inaccuracies.

The equations in the three Appendices (A, B, C) are marked with A, which perplexes the studying of the text. So for instance, in the explanation of the choice of the coefficients a_1 and a_2 (of Eq. (A.1) in App. C) it is written “ This choice of coefficients solve equation A.1 in series at the order 3”. Here are two questions: i) Eq.A.1 of which Appendix and ii) series of which parameter have the authors in mind? The answer lies in Eq.A.2, which after a careful examination of all

three Appendices is recognized as Eq. A.1 of Appendix B. The series parameter becomes clear from the term $o(x^3)$ of Eq.2.

As known from the theory at series expanding of a function $f(x)$ all independent parameters (including the function) should be scaled (dimensionless) and the variable x should be smaller 1 (the so called small parameter). The variable x ($0 < x < h/2$) in the present case is neither scaled, nor a small one. As the authors themselves have noted, the small parameter here seems to be $\cot\theta$, or at least proportional to $\cot\theta$ but all these details should be carefully obtained.

In the parabolic approximation of the bridge generatrix $r(x)$ for symmetric bridges $r(x) = r(\theta, x)$ the coefficient $a_1 = 0$, i.e. $r(x) = a_0 + a_2x^2$, with $a_2/2 = (r''(x))_{x=0}$ i.e. the generatrix curvature in its extremum. Further on, a_2 is in a simple relation with the contact angle θ , etc., etc.

Surprisingly, the expression of a_1 (see Eq.A.1, App.C) is not nullified by inserting F and θ from A.5-6 there!

At the end again about the Eq.A.2, App.C; more precisely about its informative value. It is presented in an identity (tautological) form ($0=0$) doing nothing to help the reader understand the method of determining the coefficients. The vagueness is also increased by the lack of a square root (apparently typographical error) in the second brackets. Actually (as far as I understood the procedures of the coefficients a determination), the parabolic polynomial $r(x)$ is inserted in Eq. A.1, App. B and nullifying the multiplier of each x^n term. An editorial in this sense would make it easier to read the article and eliminate any guesswork.

To summarize, the parabolic approximation is a possible one, but it is highly recommended to be presented in its most concise and transparent form.

Decision letter (RSOS-191608.R0)

08-Oct-2019

Dear Dr Tadrist,

On behalf of the Editor, I am pleased to inform you that your Manuscript RSOS-191608 entitled "Characterisation of interface properties of fluids by evaporation of a capillary bridge." has been accepted for publication in Royal Society Open Science subject to minor revision in accordance with the referee suggestions. Please find the referees' comments at the end of this email.

The reviewers and Subject Editor have recommended publication, but also suggest some minor revisions to your manuscript. Therefore, I invite you to respond to the comments and revise your manuscript.

- Ethics statement

- Data accessibility

It is a condition of publication that all supporting data are made available either as supplementary information or preferably in a suitable permanent repository. The data accessibility section should state where the article's supporting data can be accessed. This section should also include details, where possible of where to access other relevant research materials such as statistical tools, protocols, software etc can be accessed. If the data has been deposited in an external repository this section should list the database, accession number and link to the DOI

for all data from the article that has been made publicly available. Data sets that have been deposited in an external repository and have a DOI should also be appropriately cited in the manuscript and included in the reference list.

If you wish to submit your supporting data or code to Dryad (<http://datadryad.org/>), or modify your current submission to dryad, please use the following link:
<http://datadryad.org/submit?journalID=RSOS&manu=RSOS-191608>

- **Competing interests**

- **Authors' contributions**

- **Acknowledgements**

- **Funding statement**

Because the schedule for publication is very tight, it is a condition of publication that you submit the revised version of your manuscript before 17-Oct-2019. Please note that the revision deadline will expire at 00.00am on this date. If you do not think you will be able to meet this date please let me know immediately.

Best regards,

Lianne Parkhouse
Royal Society Open Science
openscience@royalsociety.org

on behalf of Professor Alban Potherat (Associate Editor) and Professor R. Kerry Rowe (Subject Editor)
openscience@royalsociety.org

Associate Editor Comments to Author (Professor Alban Potherat):

The referees are satisfied that their most important concerns have been addressed, however they recommend more clarity on the analytical calculations and suggest mentioning important aspects of the problem in the conclusion. The paper should be revised to address these points.

Reviewer comments to Author:

Reviewer: 1

Comments to the Author(s)

I am rather satisfied with this last response though it could be actually interesting to investigate much more the case of a metallic substrate whose role on evaporation kinetics was not really clarified. Ageing of the dynamic contact angle stands also as a decisive ingredient to be taken into account for any future modeling.

These two points must be clearly mentioned in the conclusion.

Reviewer: 3

Comments to the Author(s)

“Characterization of interface properties of fluids by evaporation of a capillary bridge” by L. Tadrict¹, L. Motte², O. Rahli² and L.

Here are my comments to Appendix C and to the answer of my review.

In general, Appendix C meets the requirement of an analytical derivation of Eq. (4.8), but at a carefully reading one finds still some inaccuracies.

The equations in the three Appendices (A, B, C) are marked with A, which perplexes the studying of the text. So for instance, in the explanation of the choice of the coefficients a_1 and a_2 (of Eq. (A.1) in App. C) it is written “ This choice of coefficients solve equation A.1 in series at the order 3”. Here are two questions: i) Eq.A.1 of which Appendix and ii) series of which parameter have the authors in mind? The answer lies in Eq.A.2, which after a careful examination of all three Appendices is recognized as Eq. A.1 of Appendix B. The series parameter becomes clear from the term $o(x^3)$ of Eq.2.

As known from the theory at series expanding of a function $f(x)$ all independent parameters (including the function) should be scaled (dimensionless) and the variable x should be smaller 1 (the so called small parameter). The variable x ($0 < x < h/2$) in the present case is neither scaled, nor a small one. As the authors themselves have noted, the small parameter here seems to be $\cot\theta$, or at least proportional to $\cot\theta$ but all these details should be carefully obtained.

In the parabolic approximation of the bridge generatrix $r(x)$ for symmetric bridges $r(x) = r(\theta, x)$ the coefficient $a_1 = 0$, i.e. $r(x) = a_0 + a_2 x^2$, with $a_2/2 = r''(x=0)$ i.e. the generatrix curvature in its extremum. Further on, a_2 is in a simple relation with the contact angle θ , etc., etc.

Surprisingly, the expression of a_1 (see Eq.A.1, App.C) is not nullified by inserting F and θ from A.5-6 there!

At the end again about the Eq.A.2, App.C; more precisely about its informative value. It is presented in an identity (tautological) form ($0=0$) doing nothing to help the reader understand the method of determining the coefficients. The vagueness is also increased by the lack of a square root (apparently typographical error) in the second brackets. Actually (as far as I understood the procedures of the coefficients a determination), the parabolic polynomial $r(x)$ is inserted in Eq. A.1, App. B and nullifying the multiplier of each x^n term. An editorial in this sense would make it easier to read the article and eliminate any guesswork.

To summarize, the parabolic approximation is a possible one, but it is highly recommended to be presented in its most concise and transparent form.

Author's Response to Decision Letter for (RSOS-191608.R0)

See Appendix E.

Decision letter (RSOS-191608.R1)

06-Nov-2019

Dear Dr Tadrict,

I am pleased to inform you that your manuscript entitled "Characterisation of interface properties of fluids by evaporation of a capillary bridge." is now accepted for publication in Royal Society Open Science.

on behalf of Professor Alban Potherat (Associate Editor) and R. Kerry Rowe (Subject Editor)
openscience@royalsociety.org

Appendix A

Royal Society Open Science – Letter to Reviewer

Dear Reviewer,

Thank you for your review. Please find a point-by-point response to your remarks.

1. **A bit of literature on capillary bridges is mentioned albeit very quickly since the configuration here studied, as well as the associated objectives and experiments, have also been extensively studied by a research group of Limoges Univ., both in hydrophilic and hydrophobic cases. This part of existing literature is passed over in silence in the present paper (see doi : 10.4236/ampc.2016.67017 and doi : 10.4236/ampc.2017.74009, Advances in Materials Physics and Chemistry).**

We thank the reviewer to point out the missing literature, namely the works done in Limoges Univ. We were aware of those papers, they were to be included in our manuscript. However, they do not appear in the first version of the manuscript by lack of care of our part. We are really sorry of this. The two papers are now fully in our short literature mentioned.

We added to the manuscript : *“Recently, the evaporation of a capillary bridge between two plates has been extensively explored in well controlled conditions by Portuquez et al.[11,12] for different wetting angles and air humidities. However capillary forces of evaporating liquid bridges were not directly measured with their set-up.”*

2. **One main question remains after reading the article : why should one study a capillary bridge rather than a sessile drop if the objective is to study the behaviour of an interface under the effect of evaporation ? Q0. What are the distinguishing arguments ?**

Again, we thank the reviewer to point out that our paper does not explain enough the interest of the configuration studied, especially when compared to the simple case of a sessile droplet. We directly show in the text of the manuscript the three main interest at studying the evaporation of a capillary bridge rather than a sessile droplet : *“It differs from the evaporation of a sessile drop in three main aspects. First experimentally, the control of the height of the bridge allow to start the drying with receding contact angles directly, whereas in the case of a sessile drop the contact angle may vary from advancing to receding at the beginning of the drying. Second, the control of the height of the bridge also allow to control finely the evaporation kinetics of the liquid. Finally, the capillary force might be measured which gives a direct insight on the evolution of the surface properties. This differs from the case of a sessile drop for which this quantity is not easily accessible. Those three aspects makes the capillary bridge technique more robust than the sessile drop technique to characterise the surface properties of fluids.”*

3. **Q1.** Evaporation kinetics with glass and PVC plates is found different at short time from that with aluminum plates (Fig. 4b). Surface state is briefly evoked as a possible reason for this. It must be checked by testing differentially polished aluminum supports.

The test over several aluminium substrates would surely bring new insight on the anchorage of the triple line when the bridge dries. However, we used an aluminium tape to perform the experiment. The aluminium tape is made of an aluminium foil placed over an adhesive part. The roughness of an aluminium foil is in the order of half a micron [5] and roughness smaller than that are really hard to perform. Nevertheless we plot the bridge profile and the contact angles temporal evolutions. We clearly see that at the beginning, the contact angles and the bridge profile show the stick-slip behaviour of the triple line. We added to the manuscript :*“The triple line stick-slip behaviour in the early time of the evaporation process is present both on the temporal evolution of the profile of the liquid bridge, figure 5a and in the temporal evolution of the lower wetting angle which fluctuates in the early stages of drying before decreasing more smoothly, figure 5b.”*

FIGURE 1 – (Color online) a. One-sided profile of a drying capillary bridge on an aluminium substrate. Time evolves from dark to light. Stick-slip motion is seen at the contact line of the capillary bridge. b. Upper (blue star) and lower (red diamond) wetting angles temporal evolution.

4. **Q2.** It can be highly recommended to measure the temperature distribution along the two plates so as to check that whatever the nature of the materials is (thermally conductive or not), the whole configuration can be considered as only driven by a small? Biot number. A natural question arises as to whether the boundary conditions for the thermal problem are really considered by the authors. No mention to this particular point is made in the paper since only the room temperature is given.

We thanks the reviewer to raise the question of the temperature in our set-up. It is indeed an important point that we did not discuss in our first version. However, we did not discussed it because temperature only plays little role

in our set-up. Our answer is supported by detailed works [1, 2] of one of the co-authors (Lounès Tadrìst) about the evaporation of sessile drops, the surface temperature gradients and the induced Marangoni flows.

We added to the manuscript : *“The evaporation of the liquid bridge induces temperature gradients at the liquid-air interface. Ait Saada et al. [1], for instance, showed that temperature differences at the interface of the order of 0.1°C occur during the evaporation of sessile drops of characteristic lengths 1 mm at room temperature with 40% humidity. They also showed that this result holds for conductive or insulating thick substrates. In our experiment, the effect of temperature variations is even more reduced because of 2 reasons : first the evaporation flux is reduced because it occurs in a Hele-Shaw (2-dimensional flux) and experiments were performed with a humidity of 0.65 larger than 0.4 in Ait Saada et al. [1] conditions. Second, the liquid is exchanging heat on two thick substrates.*

The importance of those flows can be estimated by the Marangoni Ma number [2],

$$\text{Ma} = \frac{(\partial\gamma/\partial T)\Delta T h}{\mu\alpha_T} \quad (1)$$

where γ is the surface tension, ΔT the temperature difference, h the characteristic length, μ the dynamic viscosity of the liquid and α_T the thermal diffusivity of the liquid. For pure water, $\partial\gamma/\partial T = 1.56 \cdot 10^{-4} \text{ kg/s}^2/\text{K}$, $\Delta T \simeq 0.1 \text{ K}$, $\mu = 1.0 \cdot 10^{-3} \text{ kg/m/s}$ and $\alpha_T = 1.43 \cdot 10^{-7} \text{ m}^2/\text{s}$ and a bridge in contact between two walls, $h = z_1/2 \simeq 0.75 \cdot 10^{-3} \text{ m}$, the Marangoni number is $\text{Ma} \simeq 80$. It is in the order of the critical Marangoni number above which the convective Marangoni flows occur. We thus do not expect strong Marangoni flow in our experiment. This was also pointed out by Xiao et al. [3] and Bouchenna et al. [2] which found the influence of the thermocapillary effect on the evaporation of a sessile droplet negligible at ambient temperature.

In the following, we will fully decouple the thermal problem from the mechanics of the capillary bridge.”

5. **C1. Fig. 5a is just a combination of two previous figures with no decisive input.**

We aimed at showing the two previous figures to have the reader understand how this final figure was built since evaporation is not the only way to obtain it. We think that the manuscript is clearer with this figure.

6. **Q4. The authors do not provide time-dependent profiles of the capillary bridge, $r(t,z)$, despite the possibility to get them from imaging. Why? These profiles reveal to be a decisive information, especially if the aim is to give an idea of the dynamic behaviour of the wetting angles and the triple contact line along the plates : stick-slip, no-slip motion ?**

The figure is now given. See answer to item 3.

7. **Q5. The Young-Laplace and Young-Duprè equations are derived, based on an energetic point of view, in order to calculate the normal**

force between the two plates and to justify the boundary condition based on wetting angles. In fact, this model is nothing but a simplified form of a normal momentum balance at the capillary bridge surface (“Interfacial Transport Phenomena”, Springer, by Slattery, Sagis & Oh). What are the assumptions which make this mechanical model decoupled from the diffusion-limited (evaporation) model? What is the value of the Bond number which allows for neglecting gravity effects?

The answer to the thermal part of this question is given in item 4. For the value of the Bond number necessary to neglect gravity effect, usually 0.1 is the standard. In this work, the bond number is 0.3 meaning that we expect some effects of the gravity on our results. The Bond number is given in the manuscript as “*The Bond number $B_o = \rho g z_1^2 / \gamma$, where ρ is the water density, compares the effect of gravity g to the effect of surface tension γ . In our experiment, $B_o \sim 0.3$ means that gravitational effects, although present, could be neglected at the first order.*”

8. **C2. Figure 6. The caption can not be related to the data points.**

This was a mistake. We missed to change the caption when the figure changed during the writing iteration process. We changed the caption of figure 6 (now figure 7) in “*Diffusion controlled evaporation of a liquid bridge. Direct comparison of the experimental evaporation debit with diffusion-controlled evaporation model, Equation (4.6).*”

9. **Q6. The diffusion-limited model for the evaporation as proposed in page 7 is a clone of the simple model proposed by Bexon & Picknett (The evaporation of sessile or pendant drops in still air, Journal of Colloid and Interface Science 61 (2) (1977) 336-350). The additional introduction of an equivalent radius r_e all along the capillary bridge is not necessary and a full expression based on a local time-dependent radius must be implemented. Of course, this makes the model non-linear but nowadays, this is no more an issue.**

Thank you for your comment. The first diffusion-limited model was first proposed by Langmuir in 1918 [4] to account for the evaporation of small spheres. The model proposed here is already non linear and we tried to keep things as simple as possible. The introduction of an equivalent radius is not a major assumption since we show that in our 2D system the kinetics is limited by the vapour diffusion.

10. **Q7. Evaporation is well-known to focus at the contact lines of sessile drops, yielding a thermally-induced Marangoni flow. What happens here with a capillary bridge? Can the authors justify why they do not consider thermally-induced Marangoni effect?**

Thank you for your comment. Please refer to item 4.

11. **Q8. The role of convection at the vicinity of the capillary bridge is not mentioned. Is it negligible? If so, why?**

The convection at the vicinity of the droplet is considered negligible because no strong temperature gradient exists in the Hele-Shaw cell. We added to the

manuscript :“ *The evaporation of the liquid bridge was made at room temperature and between two thick plates. The slow evaporation process ensures that no strong thermal gradient existed in this experiment. In those conditions, no air convection was expected.*”

Yours sincerely,

Loïc Tadrist, Ludovic Motte, Ouamar Rhali and Lounès Tadrist

REFERENCES

- [1] Ait Saada, M., Chikh, S., & Tadrist, L. (2013). **Evaporation of a sessile drop with pinned or receding contact line on a substrate with different thermophysical properties.** International journal of Heat and Mass Transfer, 58(1-2), 197-208.
- [2] Bouchenna, C., Saada, M. A., Chikh, S., & Tadrist, L. (2017). **Generalized formulation for evaporation rate and flow pattern prediction inside an evaporating pinned sessile drop.** International Journal of Heat and Mass Transfer, 109, 482-500.
- [3]Xiao, C., Zhou, L., Sun, Z., Du, X., & Yang, Y. (2016). **Near-wall fluid flow near the pinned contact line during droplet evaporation.** Experimental Thermal and Fluid Science, 72, 210-217.
- [4] Langmuir, I. (1918). **The evaporation of small spheres.** Physical review, 12(5), 368.
- [5] Le, H. R., & Sutcliffe, M. P. F. (2000). **Analysis of surface roughness of cold-rolled aluminium foil.** Wear, 244(1-2), 71-78.

Appendix B

Royal Society Open Science – Letter to Reviewer

Dear Reviewer,

Thank you for your review. Please find a point-by-point response to your remarks.

1. **The title is awkward (Capillary bridge evaporation study : a robust method to characterize the interface properties of fluids)**

Indeed, the title was a bit awkward. We changed it in “ Characterization of interface properties of fluids by evaporation of a capillary bridge”.

2. **L. 25 : It should be “key element” (not key bock) L. 29 : The whole Abstract should be more informative, and specific. English should be revised**

We changed “key block” in “key element” and rewrote the abstract to be more specific. The whole paper has been revised to improve the written English.

3. **“no noticeable flow occurs in the bridge and it can be considered at rest” - This statement needs to be qualified. There are findings available for other evaporating capillary bridges, sessile drops indicating strong pressure gradients and flows within the bridge as well, especially intense shortly prior bridge rupture (see e.g. Snoeijer and Anreotti, 2013, Yang et al., 2019). Coffee stain flows are the best known examples of such flows. Here, the safest would be to add that “in the early phases of evaporation”.**

This excellent remark made us write a full paragraph to quantify the Marangoni induced flows in our experiment. Our answer is supported by detailed works [1, 2] of one of the co-authors (Lounès Tadrist) about the evaporation of sessile drops, the surface temperature gradients and the induced Marangoni flows.

We added to the manuscript : *“The evaporation of the liquid bridge induces temperature gradients at the liquid-air interface. Ait Saada et al. [1], for instance, showed that temperature differences at the interface of the order of 0.1°C occur during the evaporation of sessile drops of characteristic lengths 1 mm at room temperature with 40% humidity. They also showed that this result holds for conductive or insulating thick substrates. In our experiment, the effect of temperature variations is even more reduced because of 2 reasons : first the evaporation flux is reduced because it occurs in a Hele-Shaw (2-dimensional flux) and experiments were performed with a humidity of 0.65 larger than 0.4 in Ait Saada et al. [1] conditions. Second, the liquid is exchanging heat on two thick substrates.*

The importance of those flows can be estimated by the Marangoni Ma number [2],

$$\text{Ma} = \frac{(\partial\gamma/\partial T)\Delta Th}{\mu\alpha_T} \quad (1)$$

where γ is the surface tension, ΔT the temperature difference, h the characteristic length, μ the dynamic viscosity of the liquid and α_T the thermal diffusivity of the liquid. For pure water, $\partial\gamma/\partial T = 1.56 \cdot 10^{-4} \text{ kg/s}^2/\text{K}$, $\Delta T \simeq 0.1 \text{ K}$, $\mu = 1.0 \cdot 10^{-3} \text{ kg/m/s}$ and $\alpha_T = 1.43 \cdot 10^{-7} \text{ m}^2/\text{s}$ and a bridge in contact between two walls, $h = z_1/2 \simeq 0.75 \cdot 10^{-3} \text{ m}$, the Marangoni number is $\text{Ma} \simeq 80$. It is in the order of the critical Marangoni number above which the convective Marangoni flows occur. We thus do not expect strong Marangoni flow in our experiment. This was also pointed out by Xiao et al. [3] and Bouchenna et al. [2] which found the influence of the thermocapillary effect on the evaporation of a sessile droplet negligible at ambient temperature.

In the following, we will fully decouple the thermal problem from the mechanics of the capillary bridge. ”

4. **Fig. 5-** would be more intuitive to use the horizontal axis along the process progress. Invert the horizontal axis, from highest to lowest volume.

Thanks for your advise. We inverted the horizontal axis of the figure. The figures now reads from left to right as time increases.

5. **31 : Bond number (not bond number)**

We changed the b to capital B in Bond number.

6. **18- how contact angle was measured ? In which distance from contact point ? Indicate what does it mean upper or lower contact angle ? Calculated capillary force is very sensitive to contact angle changes**

The contact angle was computed from the shape $r(z)$ of the liquid bridge and its spatial derivative $\dot{r}(z)$. The upper and lower contact angles refer respectively to the contact angle at location z_1 and 0.

We added the necessary precisions to our manuscript : “*The contact angle is deduced from the profile of the bridge $r(z)$ with equation (4.3) either at elevation $z = z_1$ for upper contact angle, either at elevation $z = 0$ for lower contact angle.*”

7. **33 : Should be : bridge was placed between two plates ? L 50-56**

We changed in “*The liquid bridge is placed between two plates [...]*”

8. **Figure 6 : where are a) b) and c) ?**

This was a mistake. We missed to change the caption when the figure changed during the writing iteration process. We changed the caption of figure 6 (now figure 7) in “*Diffusion controlled evaporation of a liquid bridge. Direct comparison of the experimental evaporation debit with diffusion-controlled evaporation model, Equation (4.6).*”

9. **56- 58- The sentence is not clear.**

We changed the sentence to make it clear : “*For the sake of simplicity we consider a case where gravity does not play any role, corresponding to small Bond numbers $B_o \rightarrow 0$. In those conditions, if the upper and lower substrate*

are identical, the capillary bridge should be symmetric $r(z + z_1/2) = r(-z + z_1/2)$. ”

Yours sincerely,

Loïc Tadrìst, Ludovic Motte, Ouamar Rhali and Lounès Tadrìst

REFERENCES

[1]Saada, M. A., Chikh, S., & Tadrìst, L. (2013). **Evaporation of a sessile drop with pinned or receding contact line on a substrate with different thermophysical properties.** International journal of Heat and Mass Transfer, 58(1-2), 197-208.

[2]Bouchenna, C., Saada, M. A., Chikh, S., & Tadrìst, L. (2017). **Generalized formulation for evaporation rate and flow pattern prediction inside an evaporating pinned sessile drop.** International Journal of Heat and Mass Transfer, 109, 482-500.

[3]Xiao, C., Zhou, L., Sun, Z., Du, X., & Yang, Y. (2016). **Near-wall fluid flow near the pinned contact line during droplet evaporation.** Experimental Thermal and Fluid Science, 72, 210-217.

Appendix C

Royal Society Open Science – Letter to Reviewer

Dear Reviewer,

Thank you for your review. According to your main remark, we built an analytic model to understand the role of each parameter on the capillary force.

1. **I found very interesting the experimentally established dependencies, $F \sim z^{-3/2}$ at constant volume V (Fig.4) and $F \sim 1/V$ at constant height, z (Fig.6), but they need a theoretical analysis. Taking into account that in the literature can be found analytical solution of the bridge height z and volume V as function of the same parameters (r_0, r_1) as the parameters of the force (see e.g. Capillary Bridges in Wikipedia and the literature cited there) such an analysis is only a routine procedure. Moreover, graph in Figure 6b looks more as a curve (with negative second derivative) than as a straight line. Do that analysis and the study will be a very useful and instructive paper.**

We added a full model of capillary forces to our analysis. Our model is analytic but is restrained to the range of slightly curved bridges (contact angles close to 90°). It differs from the model of Chen et al. that considers advancing and receding contact angles during the drying process. Our model shows the role of each parameter that play a role in the capillary force. The full computation of the force is given in appendix. We added to the text :

To rationalize the dependency of the capillary force F with the height of the bridge z_1 , we carried out a theoretical analysis by solving the Young-Laplace equation for slightly curved capillary bridges (with identical contact angles close to 90° , see appendix C). In this framework, we derived an analytical solution of the capillary force as a function of experimental parameters, bridge volume V , bridge height z_1 and contact angle θ ,

$$F = \gamma \sqrt{\frac{\pi V}{z_1}} + \left(\frac{2\gamma V}{z_1^2} - \frac{\pi\gamma z_1}{12} \right) \cot \theta + o(\cot^2 \theta) \quad (1)$$

The comparison between the experimental results and the theoretical ones for the three bridge volumes show the right trend and a pretty good agreement, Figure 3b. The contact angle have been fitted to $\theta_{\text{pvc}} = 80^\circ$ for the three cases which is larger than the observed value $\theta_{\text{pvc}} \simeq 60^\circ$. The scaling law highlighted experimentally of the capillary force proportional to $F \propto z_1^{-3/2}$ corresponds in fact to a transition zone between two regimes where $F \propto z_1^{-2}$ and $F \propto z_1^{-1/2}$.

This expression of the capillary force, Equation (1), has also been compared to experimental measurements of the force in Figure 6. The contact angles have been fitted using $\theta_{\text{pvc}} = 80^\circ$, $\theta_{\text{glass}} = 55^\circ$ and $\theta_{\text{aluminium}} = 75^\circ$ which are overestimating the real contact angles. The model gives the good trend but does not match completely the experimental data. This discrepancy is due to

the assumptions considered here to solve the Young Laplace equation which only consider slightly curved bridges and does not take into account neither the stick-slip phenomenon [Chen, 2013] nor the drift in contact angle during evaporation.

Appendix D

Royal Society Open Science – Letter to Reviewer

Dear Reviewer,

Thank you for your review. Please find a point-by-point response to your remarks.

1. **Considering now the modelling part of the paper, I think that the authors would gain much benefit from considering the paper by Chen et al. (<https://doi.org/10.1021/la304870h>).**

We added a full model of capillary forces to our analysis. Our model is analytic but is restrained to the range of slightly curved bridges (contact angles close to 90°). It differs from the model of Chen et al. that considers advancing and receding contact angles during the drying process. Our model, although much less accurate than chen's model, explicit the role of each parameter that play a role in the capillary force. The full computation of the force is given in appendix. We added to the text :

To rationalize the dependency of the capillary force F with the height of the bridge z_1 , we carried out a theoretical analysis by solving the Young-Laplace equation for slightly curved capillary bridges (with identical contact angles close to 90° , see appendix C). In this framework, we derived an analytical solution of the capillary force as a function of experimental parameters, bridge volume V , bridge height z_1 and contact angle θ ,

$$F = \gamma \sqrt{\frac{\pi V}{z_1}} + \left(\frac{2\gamma V}{z_1^2} - \frac{\pi\gamma z_1}{12} \right) \cot \theta + o(\cot^2 \theta) \quad (1)$$

The comparison between the experimental results and the theoretical ones for the three bridge volumes show the right trend and a pretty good agreement, Figure 3b. The contact angle have been fitted to $\theta_{\text{pvc}} = 80^\circ$ for the three cases which is larger than the observed value $\theta_{\text{pvc}} \simeq 60^\circ$. The scaling law highlighted experimentally of the capillary force proportional to $F \propto z_1^{-3/2}$ corresponds in fact to a transition zone between two regimes where $F \propto z_1^{-2}$ and $F \propto z_1^{-1/2}$.

This expression of the capillary force, Equation (1), has also been compared to experimental measurements of the force in Figure 6. The contact angles have been fitted using $\theta_{\text{pvc}} = 80^\circ$, $\theta_{\text{glass}} = 55^\circ$ and $\theta_{\text{aluminium}} = 75^\circ$ which are overestimating the real contact angles. The model gives the good trend but does not match completely the experimental data. This discrepancy is due to the assumptions considered here to solve the Young Laplace equation which only consider slightly curved bridges and does not take into account neither the stick-slip phenomenon [Chen, 2013] nor the drift in contact angle during evaporation.

We also added the reference to Chen *et al.* to our short literature.

2. **My question Q5 (first review) on the normal momentum balance at the liquid surface is not properly addressed. Mass transfers are**

commonly responsible for an additional force whose (negligible?) role must be properly made evident.

It has been difficult for us to understand what the reviewer meant by role of mass transfer on the bridge mechanics since it is not often considered in other slow rate evaporation papers. It is well known that mass transfer only play a role when the evaporation is extremely fast. Nevertheless, we made clear the role of the mass transfer on the bridge mechanics by computing the ratio of the pressure induced by mass transfer to the capillary pressure and showed that in our experiment it is negligible. We added to the manuscript :

The mass transfer of volatile liquid to gas creates an additional recoil pressure P_r through the differential vapour recoil mechanism [Palmer1981, Delhaye1974, Nikolayev2001]. Considering the effect on mass transfer, the interface jump condition differs from the usual Young-Laplace equation by considering the recoil pressure,

$$\Delta P = P_r + \gamma\mathcal{C} \quad \text{with} \quad P_r = \left(\frac{dm}{Sdt}\right)^2 \left(\frac{1}{\rho_l} - \frac{1}{\rho_g}\right) \quad (2)$$

where \mathcal{C} is the total curvature of an interface, dm/Sdt the evaporation per unit surface, ρ_l and ρ_g the density of the liquid and the gas.

With typical values $\gamma = 72 \text{ mN}$, $V \simeq 10 \mu\text{L}$, $z_1 \simeq 11 \text{ mm}$, $\rho_l = 1000 \text{ kg m}^{-3}$, $\rho_g = 1 \text{ kg m}^{-3}$, and an evaporation time $t_e \simeq 7000 \text{ s}$ we can evaluate the ratio of the recoil pressure to the Laplace pressure $P_r/\gamma\mathcal{C}$. Using the following approximations, $\mathcal{C} \simeq 2/z_1 =$ and $dm/Sdt \simeq \rho_l\sqrt{V}/2\pi\sqrt{z_1}t_e$, we obtain

$$\left|\frac{P_r}{\gamma\mathcal{C}}\right| = \frac{\rho_l^2 V}{8\pi^2 \rho_v t_e^2 \gamma} \sim 3 \cdot 10^{-11} \ll 1 \quad (3)$$

We can thus safely neglect the effect of mass transfer on the bridge mechanics.

Appendix E

Royal Society Open Science – Response letter

Dear Editor,

We thank you for accepting our paper to be published in the journal of the Royal Society Open Science (RSOS). Some minor revisions, mainly editing of the appendix, were requested by one reviewer. We made those small changes. We think that our paper is now ready for publication in RSOS.

With many thanks for your careful consideration,

Loïc Tadrist, on the behalf of the authors.